# Rapid Microstructure Homogenization of a Laser Melting Deposition Additive Manufactured Ti-6.5Al-3.5Mo-1.5Zr-0.3Si Alloy by Electropulsing

**DOI:** 10.3390/ma15207103

**Published:** 2022-10-13

**Authors:** Dandan Ben, Huajie Yang, Jiabao Gao, Bingyu Yang, Yu’ang Dong, Xiangyu Liu, Xuegang Wang, Qiqiang Duan, Peng Zhang, Zhefeng Zhang

**Affiliations:** 1Shi-Changxu Innovation Center for Advanced Materials, Institute of Metal Research, Chinese Academy of Sciences, Shenyang 110016, China; 2Department of Materials Science and Engineering, University of Science and Technology of China, Shenyang 110016, China; 3Shenyang Zhongke Raycham Technology Co., Ltd., Shenyang 110200, China

**Keywords:** titanium alloy, laser melting deposition, electropulsing, microstructure, homogenization

## Abstract

The typical microstructure of the laser melting deposition (LMD) additive-manufactured Ti-6.5Al-3.5Mo-1.5Zr-0.3Si alloy (TC11) contains the heat-affected bands (HABs), the narrow bands (NBs) and the melting pools (MPs) that formed due to the reheating and superheating effects during the layer-by-layer manufacturing process. Characterization results indicated that the coarse primary α lath (α_p_) and transformed β (β_t_) structures were located in the HABs, while the fine basketweave structure was formed inside the MPs. The rapid modifications of microstructure and tensile properties of the LMD-TC11 via electropulsing treatment (EPT) were investigated. The initial heterogeneous microstructure transformed into a complete basketweave structure and the HABs vanished after EPT. Thus, a more homogeneous microstructure was achieved in the EPT sample. The ultrafast microstructural changes were mainly attributed to the solid state phase transformation during electropulsing. The tensile properties of the sample were basically stable, except that the yield strength decreased as EPT voltage increased. This study suggests that EPT could be a promising method to modify the microstructure and mechanical properties of the additive-manufactured alloys in a very short time.

## 1. Introduction

Titanium alloys have been widely utilized in the aerospace industry, marine industry, biomedicine, chemical industry and other industrial fields due to their excellent specific strength, fracture toughness, temperature resistance and corrosion resistance [1,2,3]. Nowadays, conventional processing methods such as casting, rolling, forging, machining and other methods are used to fabricate the titanium alloy components [1]. However, the high melting point, chemical activity, narrow processing temperature and small deformation coefficient of titanium alloys arouse high cost and manufacturing difficulties, resulting in a high buy-to-fly ratio, energy consumption and material waste [4,5]. Additive manufacturing (AM) technology, a layer-by-layer manufacturing technology based on sliced data from three-dimensional CAD models, has been developed as a material-saving and design-free method in recent several decades [6,7]. AM technologies could be clarified as directed energy deposition (DED) and powder bed fusion (PBF) manufacturing processes according to ASTM [8]. DED could be utilized to fabricate large-scale components with high efficiency and flexibility, while PBF could manufacture fine components with excellent mechanical properties. Due to the unique layer-by-layer manufacturing method, microstructural anisotropy and inhomogeneity become essential issues in additively manufactured alloys [4]; thus, it is necessary to develop the post-treatment to realize microstructural homogenization.

Ti-6.5Al-3.5Mo-1.5Zr-0.3Si (TC11) alloy is an α + β type heat-resistant titanium alloy characterized by good comprehensive properties at room and elevated temperatures, such as high strength, excellent thermal stability and good creep properties [9]. As reported [10,11,12], the yield strength, ultimate strength and elongation of the forged TC11 alloy are 968–1020 MPa, 1043–1181 MPa, and 9.7–16.5%, respectively, after various heat treatment histories. This alloy is usually used in compressor blades and discs in aero-engines [13,14]. Laser melting deposition (LMD) technology, acting as one typical DED method, has been applied in the manufacturing of TC11 alloy, overcoming the processing difficulties of traditional methods. Zhu et al. [12,15] investigated the grain morphologies, microstructural features and mechanical properties, exploring the alternate arrangement of columnar grains and equiaxed grains, the interlayer bands (ILBs) that were attributed to the superheating effect and the heat-affected bands (HABs) that were caused by re-heat treatment processes. The LMD TC11 alloy shows typical microstructural morphologies of large columnar grains and MPs, which could arouse anisotropy of microstructure and mechanical properties. Therefore, it is necessary to take out post-treatment methods to optimize the microstructure and mechanical properties of LMD TC11 samples. Zhu et al. [16] have investigated the effects of heat treatment on the microstructures of LMD TC11 alloy and found that the α_p_ and β_t_ structures were not homogeneous after the α + β heat treatment and the β heat treatment. Different from traditional heat treatment, electropulsing treatment (EPT) was developed to tailor the microstructure and mechanical properties of metal alloys in an instant way. So far, EPT has been applied to realize recrystallization [17,18], phase transformation [19,20], precipitation configuration [21,22], inclusion elimination [23] and crack healing [24,25,26] based on the athermal and thermal effects induced by the pulsed current in the metallic materials. Electropulsing has been conducted on the TC11 alloy [27] and the study was focused on the phase content and texture distribution evolutions. Recently, several studies have paid attention to the investigations of EPT and additive manufactured metals. Noell et al. [28] studied the microstructural modification in as-fabricated SLM 316 L stainless steel and AlSi10Mg parts under electrical pulses, and the chemical microsegregation could be eliminated. Waryoba et al. [29] found that athermal effects contributed to electro-strengthening and residual stress could be reduced in the AM Ti6Al4V alloy. Recent research on the SLM Ti6Al4V alloy reported that rapid solid-state phase transformation could be induced by EPT, realizing grain morphology transition from columnar grains to equiaxed grains [30].

In this study, the high energy density EPT was implemented on the as-deposited LMD-TC11 alloy, and the microstructural modification and mechanical property evolutions are here presented.

## 2. Experimental

The TC11 alloy was fabricated using LMD technology with laser power of 2.4 kW, laser beam diameter of 4.5 mm, laser scanning speed of 12 mm/s and powder feed rate of 340 g/h; a layer thickness of 0.7 mm and overlapping distance of 2.2 mm were chosen for the starting material. The LMD process and cuboid-shaped EPT sample with a dimension of 50 × 6.5 × 2.5 mm^3^ are shown in Figure 1. The axial direction of the EPT sample was perpendicular to the building direction. The pulsed current was provided by 10 sets of capacitors. The capacitors were charged to the set voltage and then discharged instantly; thus, an attenuated pulsed current was obtained. The detected pulsed time through an oscilloscope was about 400 ns. The EPT voltages were selected as 6 kV, 7 kV, 8 kV and 9 kV at ambient temperature and the EPT samples were referred to as EPT6, EPT7, EPT8 and EPT9 samples. The original TC11 was named as EPT0 sample. The samples for the stereoscope, optical microscopy (OM) and scanning electron microscopy (SEM) observations were mechanically polished with 50 nm colloidal silica and then chemically etched in an etching solution of 2% hydrofluoric acid, 12% nitric acid and 86% distilled water. The OM observations were taken on OLYMPUS DP73 equipment and the SEM observations were carried out via FEI Inspect F50 SEM equipment. The samples for EBSD analyses were electropolished in the mixed solution of perchloric acid:N-butanol:methanol = 5:35:60 at 30 V under −30 °C for 1 min. The EBSD analyses were carried out on a Zeiss Sigma 500 SEM equipped with Oxford Instrument Aztec software at an accelerating voltage of 20 kV with a step size of 0.15 μm and 0.05 μm. The grain sizes were measured through the grain detection program in the Channel 5 EBSD analysis software. The narrow axis, which is referred to as lath width, was used to describe the grain morphology. The tensile tests were accomplished on an Instron 5982 testing equipment at a strain rate of 10^−3^/s. During tension, the tensile strain could be detected and measured through the strain sensor in the testing machine and by an external extensometer. The latter one provides strain data with more accuracy but endures higher risk at deformation strains close to the fracture. Thus, in the current study, an external extensometer was used to measure the strain until 2% to obtain the yield strength. Thereafter, the strains from 2% to failure were measured by the strain sensor in the testing machine.

## 3. Results and Discussion

As well known, powder and laser moved simultaneously in the LMD process, which is also referred to as laser coaxial powder-feeding technology [31]. Owing to the comprehensive effects of high laser energy input and scanning path overlaps, powders would experience a complex thermal history of melting, solidification and reheating. Figure 2 presents the stereoscope, OM and SEM images of the as-deposited LMD-TC11 sample. As seen from the yoz plane, there are notable large melting pools (MP) with a height of 1.1 mm and width of 4.4 mm caused by laser input and scanning strategy. Columnar grains with an average width of about 0.5 mm grow through several layers driven by epitaxial growth and the growth direction is perpendicular to the melting pools. The equiaxed-like grains shown on the xoy plane are the cross-sections of the columnar grains. On the side of the xoz plane, there are layer strips with an average width of 0.6 mm parallel to the laser scanning tracks. Different from the small-scale MPs of SLM parts [32], the MPs in the LMD samples are relatively larger, containing two types of distinct MPs characterized by wide bands and narrow bands respectively, as shown in Figure 2b. The enlarged microstructural image of the wide band is presented in Figure 2c and a special bimodal structure consisting of coarse primary α (α_p_) laths and fine β transformed (β_t_) microstructure could be found. However, the narrow bands marked by an indentation are characterized by a fine basketweave microstructure that is virtually identical to that inside MPs formed due to rapid cooling rate (Figure 2d,e). To our knowledge, wide bands and narrow bands are induced by layer-by-layer building behaviors, and wide bands are commonly referred to as the heat-affected bands (HABs). As reported by Zhu et al. [15], the manufacturing process of the *N* + 3 layer could affect the *N* layer to form the α_p_ and β_t_ structures in the HABs. Concerning the narrow bands, these bands remained after annealing in the single β region and they could be attributed to the chemical composition homogenization difference caused by the superheating effect in the layer-by-layer manufacturing process [16]. The grains in the narrow band, the wide band and inside the MP were also characterized using EBSD and EBSD results are shown in Figure 3b,e,h. It could be found that the basketweave morphologies existed in the narrow band and inside the MP, while in the wide band, coarse α_p_ grains become much more common, which is consistent with the SEM images. According to statistical analysis, the average width of α laths in the wide band is 0.61 μm, higher than that in the narrow band and inside the MP, which are 0.52 μm and 0.54 μm, respectively. Moreover, the t-test was used to compare the mean of the lath widths to verify that the difference in lath widths is statistically significant.

After being treated with different EPT voltages, OM observations were taken out and the OM images are shown in Figure 4. It could be found that when the EPT voltage was chosen below 8 kV, as seen from Figure 4a,b, the yoz planes are composed of MPs with wide bands and narrow bands, almost the same as that in the as-deposited samples. Interestingly, wide bands disappeared as the EPT voltage reached 8 kV and 9 kV (Figure 4c,d). The columnar grains still existed after EPT, which was different from the equiaxed grains found in the SLM Ti6Al4V alloy after being treated at 8 kV and 9 kV [30]. As reported by Gao et al. [30], the average width of the columnar grains in the SLM Ti6Al4V alloy was 100 μm and the columnar grains transformed to the equiaxed grains after being treated with 8 kV. The difference in the columnar grain evolutions in the LMD and SLM Ti6Al4V alloys may be caused by the large size (several hundred microns) of columnar grains in the LMD samples, as seen in Figure 2a.

Detailed SEM observations were carried out to further analyze the microstructures. The typical microstructure in wide bands was used to monitor the microstructural evolutions. After being treated with EPT voltages of 6 kV and 7 kV, as shown in Figure 5a,b, the wide bands present a similar bimodal structure as that in the as-deposited sample. After being treated with EPT voltages of 8 kV and 9 kV, there are no obvious wide band MPs (Figure 4) and the microstructural morphologies become more uniform, as presented in Figure 5c,d. It could be seen that there are more homogeneous α_p_ + β_t_ basketweave structures in the EPT-8 and EPT-9 samples compared with that in the as-deposited samples. The area fractions of the β_t_ structure in the as-deposited and EPT samples were calculated using the SEM images in Figure 2b and Figure 5. The area fraction values of β_t_ are 10.5%, 8.6%, 7.3%, 18.8% and 16.0% for the EPT0, EPT6, EPT7, EPT8 and EPT9 samples, respectively. It could be seen from Figure 6 that the β_t_ content presents an increasing trend as EPT voltage increases.

Moreover, EBSD was used to describe the grain morphologies of EPT7, EPT8 and EPT9 samples. The inverse pole figures (IPFs) and α lath width distributions are presented in Figure 7, Figure 8 and Figure 9. For the EPT7 sample, the microstructures in the narrow band, wide band and inside the MP are nearly the same as the EPT0 sample. According to the EBSD data, the α lath widths in the narrow band, wide band and inside the MP are 0.55 μm, 0.63 μm and 0.53 μm. We found that the wide band is characterized with wider α lath in the EPT7 sample. For the EPT8 and EPT9 samples, the microstructure in the narrow band presents almost the same features as that inside the MP, corresponding with the SEM images shown in Figure 5. The α lath widths in the narrow band and inside the MP are 0.47 μm and 0.50 μm for the EPT8 sample, and 0.48 μm and 0.48 μm for the EPT9 sample. From above, it could be concluded that the grain morphologies and grain sizes become homogeneous in the TC11 sample after being treated with 8 kV and 9 kV.

The tensile engineering strain–stress curves of the TC11 samples before and after EPT are shown in Figure 10a. To further analyze the tensile properties of the samples, yield strength and ultimate tensile strength versus EPT voltages are plotted in Figure 10b. The yield strength and ultimate strength of the original EPT0 sample are 879 MPa and 953 MPa. The yield strengths of the EPT6 and EPT7 samples are 862 MPa and 872 MPa, similar to that of the LMD sample. While the yield strengths of the EPT8 and EPT9 samples are 726 MPa and 771 MPa, exhibiting a decreasing trend generally, which could be attributed to the decrease of α_p_ and the increase of β_t_ [11], the EPT8 sample possesses the lowest yield strength, which is corresponding to the highest area fraction value of β_t_. The strength and ductility of the EPT8 samples are consistent with the general strength–ductility trade-off relationship, which means the higher the ductility the lower the strength. In the additively manufactured sample, besides microstructural evolutions, the printing defects could also affect the ductility significantly, arousing the instability in tensile ductility. Thus, the change in ductility is more complicated, and it needs to be further investigated.

From above, after being treated with low EPT voltages of 6 kV and 7 kV, the microstructure was seemingly unchanged. While the EPT voltages reached 8 kV and 9 kV, the heterogeneous microstructure was transformed to a more homogeneous α + β basketweave structure, as presented in Figure 4 and Figure 5. It could be inferred that rapid α → β phase transformation took place during EPT, resulting in microstructural homogenization and the disappearance of the wide bands. EPT-induced phase transformation has been reported in steels [19], titanium alloys [30], Cu-Zn alloy [33] and so on [20]. From the aspect of energy, the relationship between the total energy change in the current-carrying system, $\Delta G^{\mathrm{EPT}}$, and the energy change induced by electric current, $\Delta G^{e}$, could be expressed as [34]
(1)$$\Delta G^{\mathrm{EPT}}=\Delta G^{0}+\Delta G^{e}$$
(2)$$\Delta G^{e}=\mu_{0}g\xi \left(\sigma_{1},\sigma_{2}\right)j^{2}\Delta V$$
where $\mu_{0}$ is the magnetic susceptibility in vacuum, g is a positive geometric factor, j is the current density, ΔV is the volume of a nucleus, $\sigma_{1}$ and $\sigma_{2}$ are the electric conductivities of the α phase and β phase, $\xi \left(\sigma_{1},\sigma_{2}\right)$ is described as $\xi \left(\sigma_{1},\sigma_{2}\right)=\left(\sigma_{1}-\sigma_{2}\right)/\left(\sigma_{1}+\sigma_{2}\right)$, and $\Delta G^{0}$ is the energy change of the current-free system. According to the previous literature [35,36], $\sigma_{1}<\sigma_{2}$ and $\Delta G^{e}<0$ could be obtained during the α → β phase transformation process. Thus, the current-carrying system would make it easier to realize phase transformation in a short time compared with the current-free system due to the decrease in transformation barrier induced by electropulsing.

Additionally, the α → β phase transformation in α + β titanium alloy is controlled by atomic diffusion [37]. As reported by previous literature [38,39], the atomic diffusion could be enhanced by thermal and athermal effects under EPT. The total atomic flux J consisted of the atomic flux induced by the thermal effect $J_{t}$ and the atomic flus induced by the athermal effect $J_{a}$. The relationships are described as Equations (3)–(5):(3)$$J=J_{t}+J_{a}$$
(4)$$J_{t}=\frac{D_{1}}{kT}\tau \Omega$$
(5)$$J_{a}=\frac{D_{1}}{kT}\left(K_{\mathrm{ew}}\Omega J_{m}+N_{1}\rho eZ_{1}^{*}J_{m}\right)$$
where $D_{1}$ is the lattice diffusion coefficient, k is the Boltzmann constant, T is the temperature, τ is the external stress, Ω is the atom volume, $K_{\mathrm{ew}}$ is the coefficient of electron external stress, $J_{m}$ is the amplitude of the current density of electropulsing, $N_{1}$ is the number of lattice atoms per unit volume, ρ is the electrical resistivity, and $eZ_{1}^{*}$ is the effective charge of lattice atoms. It could be seen that the athermal effect, which is commonly referred to as electron wind force, contributes to the movement of atoms and the thermal effect also affects atom diffusion, as reflected by Equation (3).

To further illustrate the microstructural evolution during EPT, the schematic diagram is presented in Figure 11. The as-deposited sample is characterized by a special bimodal structure in the HABs and fine basketweave microstructure inside the MPs, as shown in Figure 11a. The heterogeneous microstructure transforms into uniform fine basketweave microstructure and HABs disappear owing to the rapid phase transformation under the appropriate EPT voltage (Figure 11b). However, the narrow bands remain unchanged after EPT.

## 4. Conclusions

The microstructure and tensile properties of the LMD-TC11 alloy before and after EPT were investigated. Heterogeneous microstructures containing the columnar grains with α + β bimodal and basketweave structures were found in the as-deposited alloy due to the reheating in the layer-by-layer manufacturing process. As EPT voltage increased, the HABs vanished and the α + β bimodal structure changed to a more homogeneous fine basketweave microstructure. The tensile properties of the samples were basically stable, except the yield strength decreases ~17% and ~12% for the EPT8 and EPT9 samples due to the decrease of the primary α (α_p_) phase and increase of transformed β (β_t_) structure. The microstructural homogenization phenomenon was attributed to the EPT-induced rapid solid-state phase transformation, which may be beneficial to the fatigue properties of the material.

## Figures and Tables

**Figure 1 materials-15-07103-f001:**
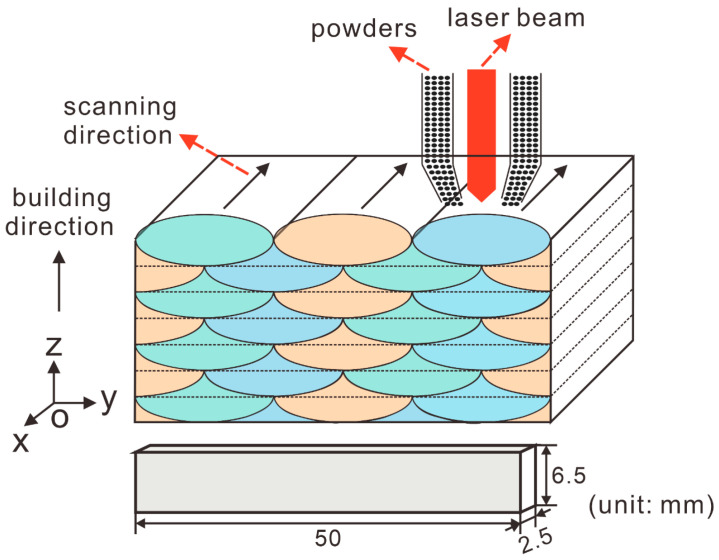
Schematic diagrams of LMD process of TC11 alloy and EPT sampling on the yoz plane.

**Figure 2 materials-15-07103-f002:**
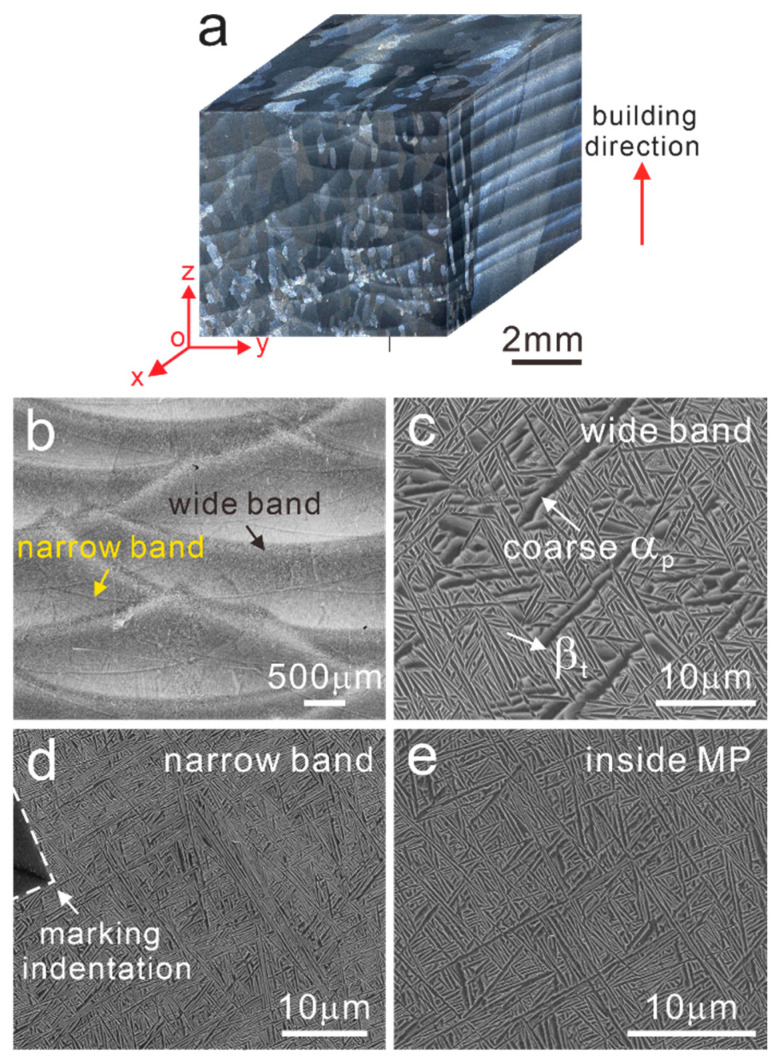
(**a**) Stereoscope optical image, (**b**) OM image and (**c**–**e**) SEM images of the MP in the as-deposited TC11 sample.

**Figure 3 materials-15-07103-f003:**
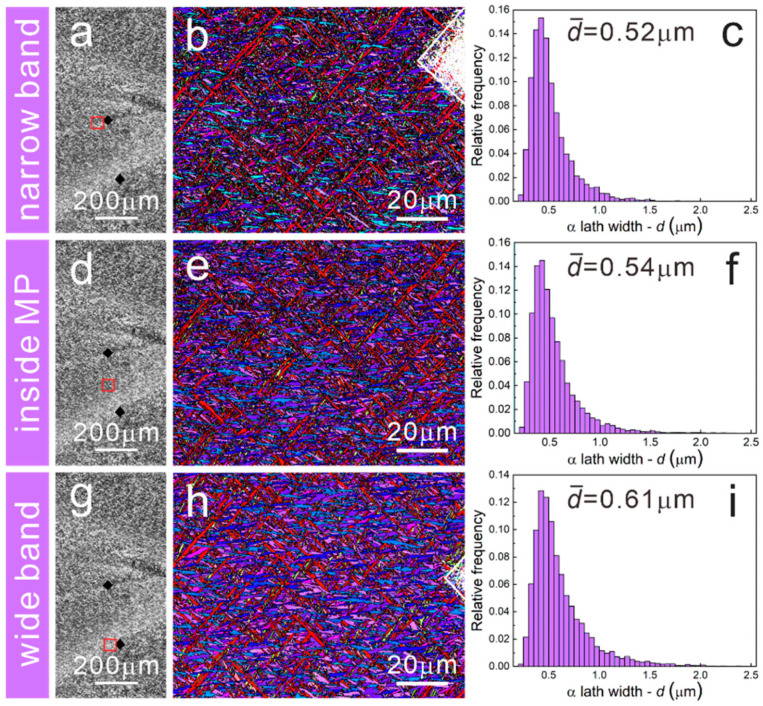
OM images and corresponding inverse pole figures (IPFs) of the EPT0 sample: (**a**–**c**) wide band, (**d**–**f**) narrow band, (**g**–**i**) inside the MP. (The black diamonds are the marking indentations of wide band and narrow band, and the red square is the EBSD scanning region.)

**Figure 4 materials-15-07103-f004:**
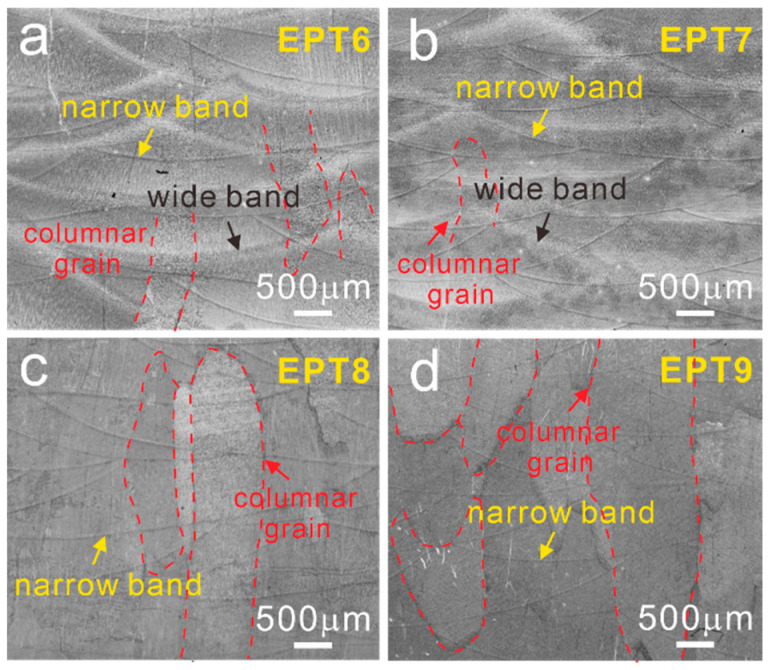
OM images of the wide bands and narrow bands in TC11 samples after being treated with different EPT voltages: (**a**) EPT6; (**b**) EPT7; (**c**) EPT8; (**d**) EPT9.

**Figure 5 materials-15-07103-f005:**
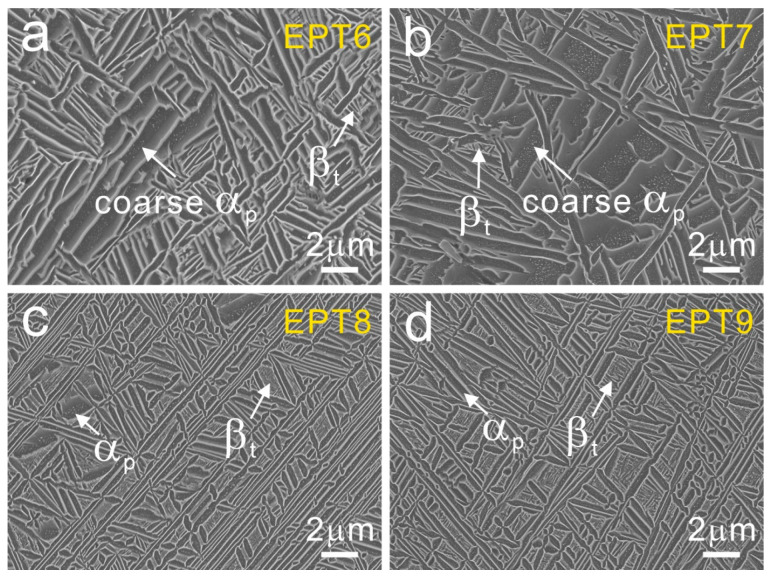
Evolutions of the bimodal structure inside wide bands in EPT samples: (**a**) EPT6, (**b**) EPT7, (**c**) EPT8, (**d**) EPT9.

**Figure 6 materials-15-07103-f006:**
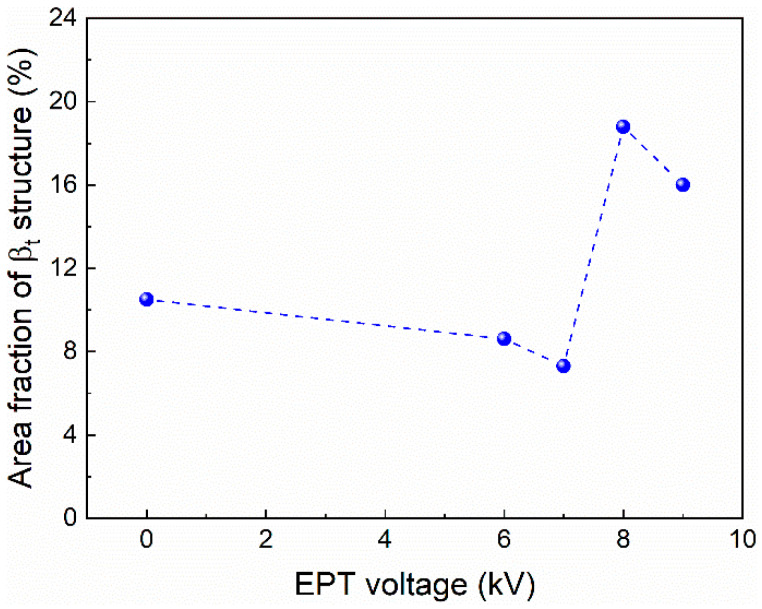
Area fractions of β_t_ structure calculated from SEM images in EPT0, EPT6, EPT7, EPT8 and EPT9 samples.

**Figure 7 materials-15-07103-f007:**
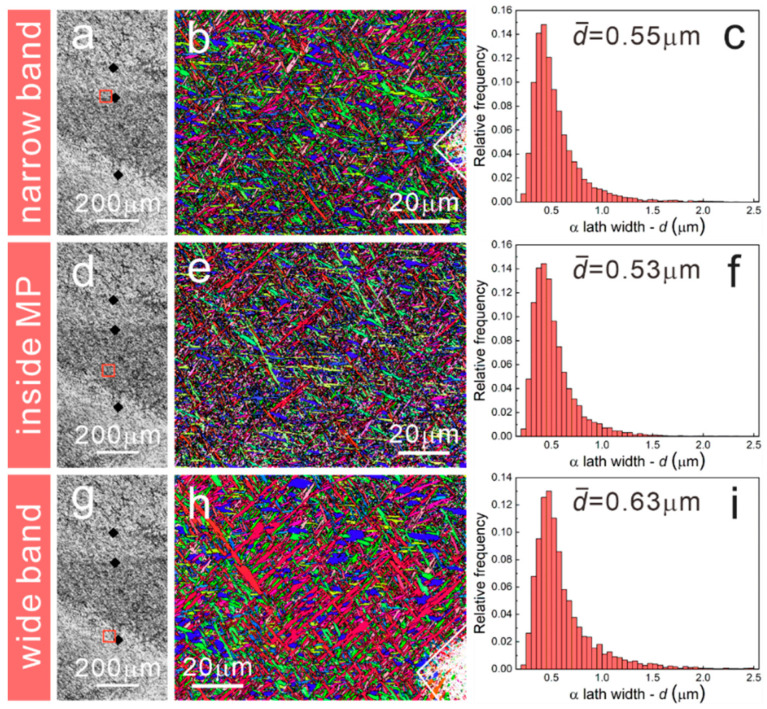
OM images and corresponding inverse pole figures (IPFs), α lath width distribution of the EPT7 sample: (**a**–**c**) narrow band, (**d**–**f**) inside the MP, (**g**–**i**) wide band. (The black diamonds are the marking indentations of wide band and narrow band, and the red square is the EBSD scanning region.)

**Figure 8 materials-15-07103-f008:**
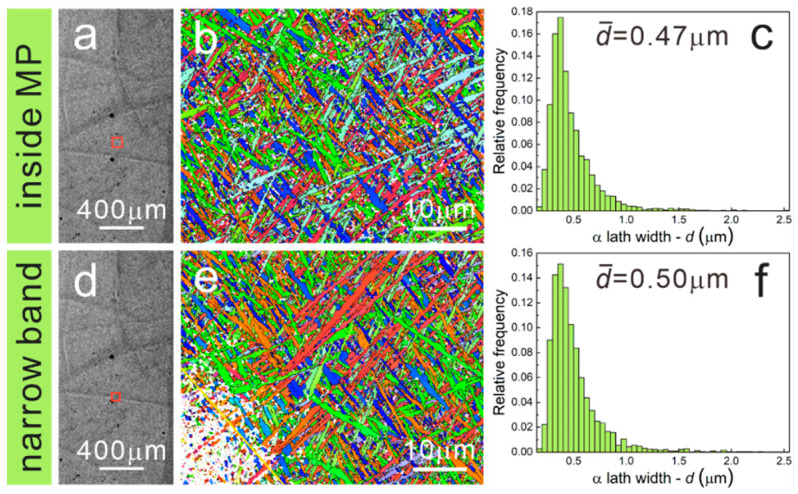
OM images and corresponding inverse pole figures (IPFs), α lath width distribution of the EPT8 sample: (**a**–**c**) narrow band, (**d**–**f**) inside the MP. (The black diamonds are the marking indentations of wide bands, and the red square is the EBSD scanning region.)

**Figure 9 materials-15-07103-f009:**
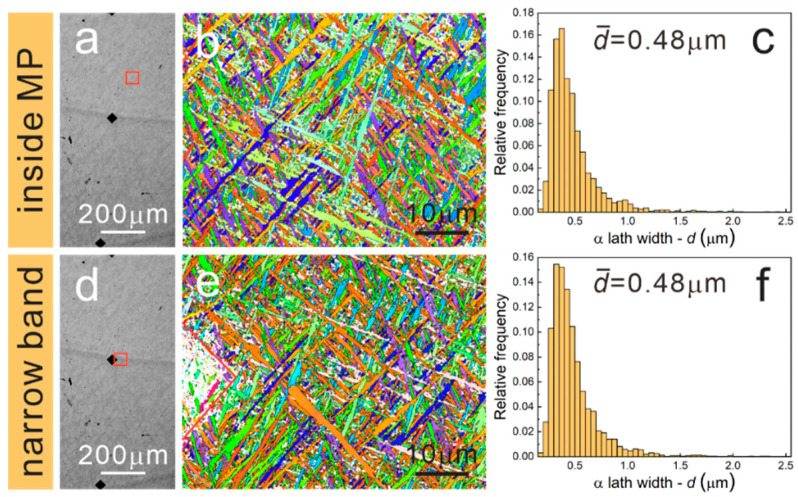
OM images and corresponding inverse pole figures (IPFs), α lath width distribution of the EPT9 sample: (**a**–**c**) narrow band, (**d**–**f**) inside the MP. (The black diamonds are the marking indentations of wide bands, and the red square is the EBSD scanning region).

**Figure 10 materials-15-07103-f010:**
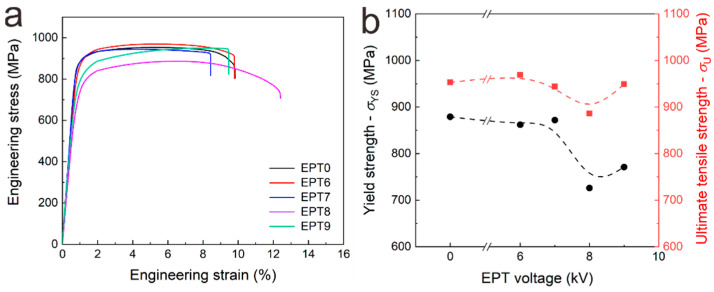
Tensile properties of the as-deposited and EPT samples. (**a**) Engineering stress–strain curves; (**b**) yield strength and ultimate tensile strength versus EPT voltages.

**Figure 11 materials-15-07103-f011:**
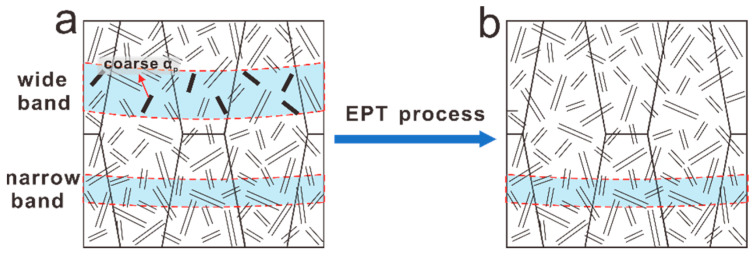
Schematic diagram of microstructural evolutions: (**a**) as-deposited sample, (**b**) EPT sample.

## Data Availability

The data used in this study are available from the corresponding authors upon request.

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
