# Peer review of "Rapid Microstructure Homogenization of a Laser Melting Deposition Additive Manufactured Ti-6.5Al-3.5Mo-1.5Zr-0.3Si Alloy by Electropulsing"

_materials, 2022, doi:10.3390/ma15207103_

Round 1

Reviewer 1 Report

The authors investigated the effect on microstructure and mechanical properties of electropulsing. Overall, the paper is of interest and the authors do a reasonable job of assessing the effects of electropulsing on these effects. However, the claims made by the authors must be supported by data. In numerous cases, claims about the effect of electropulsing on microstructure were made with no supporting data. Please add these data.

The introduction provides too little information on the TC11 alloy. What are typical mechanical properties? What are the effects on performance of ILBs and HABs?

Methods provide far too little information on how electropulsing was performed. Information on the number of pulses applied, the time between pulses, the current applied during each pulse, and the temperature rise in the samples during pulses are necessary. Additionally, there is very little information in the methods about how mechanical testing was performed, e.g. how was strain measured? Lastly, please provide more information about how EBSD data were collected. What were the accelerating voltage and the step size?

Inconsistent terminology is used to refer to the figures, e.g. Fig. 1 but Figure 2. Please either use Fig. or Figure to refer to figures.

Please add a label to Figure 2 showing the build direction.

Line 119: Please provide more detail on what you mean by the fourth layer

Line 127: there are slight differences in the lath widths but are these statistically significant? A simple t-test would provide some insight into this. Additionally, how do these findings compare to prior studies of TC11?

Line 146-147: It is impossible to tell from the OM images how the columnar grains are evolving. Please provide EBSD data to evaluate evolution of grain structure. There is no evidence from OM images that grains are evolving.

Please use consistent image scale in Figures 6-8, it is challenging to differentiate between the EBSD data in Fig 6 versus Fig 7 due to the 2X difference in image scale.

Line 173: Aside from removing wide bands, electropulsing appears to have no affect on the distribution of laths. Thus, it is not clear why the authors conclude that grain morphologies and sizes become homogenous after electropulsing. No measurements of grain size or morphology are provided, only alpha-lath spacing. Please remove this conclusion or provide supporting evidence.

Figure 9: please plot strength versus EPT as points rather than a bar chart. Bar charts are very challenging to read. Additionally, please comment on the significant change in ductility for the EPT8 sample and please provide ductility as a function of EPT

Line 226-228: No measurements of thermal inputs are provided. Thus, it cannot be concluded that the electron wind force contributed to the change.

Conclusions: the authors conclude that there is a significant alpha to Beta phase transformation induced by electropulsing, but there are no measurements of alpha and beta percentages as a function of electropulsing. These can easily be acquired using EBSD data. Please provide these to support the conclusions.

Author Response

Thank you very much for your precious time and efforts on our manuscript (materials-1903044) entitled “Rapid microstructure homogenization of a laser melting deposition additive manufactured Ti-6.5Al-3.5Mo-1.5Zr-0.3Si alloy by electropulsing” for publication in Materials. We appreciate very much for the kindness of the referee, who gave us valuable comments to improve this manuscript. We have made corrections according to the comments. The main corrections are listed in the detailed response document.

Reviewer 2 Report

The authors should enrich the literature review with extra works. The literature should also be more diversified. 

There are some similar works like Microstructural modification of additively manufactured metals by electropulsing by Noell et al / the added value should be backed up with respect to those

Link to digital twins should be made i.e. through

 1)  Stavropoulos, P., Papacharalampopoulos, A., Michail, C. K., & Chryssolouris, G. (2021). Robust additive manufacturing performance through a control oriented digital twin. Metals, 11(5), 708.

The method is not clear, i.e. EPT is not fully described.

Results are not given comparatively in a comparative way.

Results should be extended to different configurations of process parameters.

Conclusions should be extended

Author Response

Thank you very much for your precious time and efforts on our manuscript (materials-1903044) entitled “Rapid microstructure homogenization of a laser melting deposition additive manufactured Ti-6.5Al-3.5Mo-1.5Zr-0.3Si alloy by electropulsing” for publication in Materials. We appreciate very much for the kindness of the referee, who gave us valuable comments to improve this manuscript. We have made corrections according to the comments.

Round 2

Reviewer 1 Report

Introduction – Thank you for updating the introduction, but specific information on mechanical properties of this material remain absent. Please provide specific information – what are the ultimate strength, yield strength, & ductility measured by others?

Methods – I’m confused by how strain was measured. It is specified that strain was measured using an extensometer until 2%. How was strain measured thereafter? Please also provide information on how grain size was measured. Multiple methods exist, e.g. lineal intercept method, please specify.

Line 133 – please provide specifics on what statistical analysis was performed, e.g. the t-test was used to compare the mean of the distributions of the lath widths.

In response to question 6, the authors write “A prior study on the TC11 alloy…” Please add this text to the manuscript.

In response to question 7, the authors state that acquiring sufficient statistics from EBSD data would be too time-consuming for the relatively large grains observed in the study. While this may be true, I remain unconvinced by the OM images as the etch does not clearly reveal grain structure. Moreover, the statistics for grain measurements from OM images appear to be very limited, perhaps 10 grains. Please either provide clear evidence that electropulsing is changing grain size and shape or remove this conclusion, as it is not supported by the data provided.

In response to question 10, thank you for updating Figure 10. Please include your test “The decrease in strength…” in the body of the manuscript, as it will be valuable to readers.
